# Temperature dependence laser-assist scattering and its impact on efficiency of PEMFC

**Saddam Husain Dhobi**[1]*, **Kishori Yadav**[2], **Suresh Prasad Gupta**[2]*,
**Jeevan Jyoti Nakarmi**[1], **Ajay Kumar Jha**[3]

**1** Central Department of Physics, Tribhuvan University, Kirtipur, Nepal, **2** Department of Physics, Patan Multiple Campus, Tribhuvan University, Lalitpur, Nepal, **3** Department of Mechanical and Advance Engineering, Institute of Engineering, Pulchowk Campus, Tribhuvan University, Lalitpur, Nepal

* saddam@ran.edu.np (SHD); guptasiryphy@gmail.com (SPG)

## Abstract

Proton Exchange Membrane Fuel Cells (PEMFCs) is one of the best promising clean technologies in future. Numerous research activities are going on regarding to stability and thermal management of PEMFC. This reserch aim to study the scattering dynamic inside PEMFC in self-generated heat, laser field and scattering particles. To fulfil this objective first, authors developed theoretical model and then verified some parameters of theoretical model with experimental methods. For theoretical model authors formulated transition matrix using thermal Volkov wave function and thermal potential of hydrogen to study scattering dynamic. For experimental method, authors developed a PEMFC prototypes and applied diffident condition (heat and laser) to observed the data for verification of theoretical model. The developed differential cross section (DCS) model shows that with increasing temperature DCS increase theoretically and experimentally found that increasing in temperature decreasing in voltage. So, the DCS increasing with decreasing in voltage which is verified both theoretically and experimentally. In addition, we also observed that DCS effect by different parameters of PEMFC like charge transfer, charge density, efficiency, voltage, activation potential etc. and scattering parameters like momentum, scattering angle, incidence energy, distance separation etc. This finding help both academic field and non-academic field like scanning tunneling microscopy, laser-induced fluorescence, quantum computing and nanophotonic sensors perform. The finding finds that the supply temperature negatively influences PEMFC performance, which is attributed to higher particles' resistance and entropy, hence indicating how stringent is the requirement for an accurate thermal management for enhancing fuel cell efficiency.

## Introduction

PEMFC is an energy conversion system with inlets of hydrogen and oxygen by electrochemical reaction for power generation. During this process the hydrogen

**Data availability statement:** Data is fully open and available on : https://www.researchgate.net/publication/398512317_Date_set.

**Funding:** UGC-Nepal.

**Competing interests:** Authors has no competing interest.

molecules at the anode are ionized into protons and electrons. The protons traverse a proton exchange membrane and the electrons are circulated through an external circuit to produce electricity. At the cathode, recombination of protons and electrons with oxygen producing water [1]. This process is shown on the left part of Fig 1, and on right part it depicts scattering near anode electrode and surface [2]. Scattering is the process in which atomic, subatomic, or molecular particles are deflected or deviated from their original path as a result of the collision/exchange with other particles such as atoms, molecules, electrons, and photons. The one of the most important concepts applied to quantify this scattering dynamics is the DCS, which expresses how likely it is for a certain amount of particles to be scattered into an solid angle. The calculation of angular scatter distributions and insight into underlying interaction mechanisms. In the case of PEMFC, this scattering is at a catalyst-coated anode surface where electrons and protons are created. These charged particles in turn generate electric field, which may disrupt a uniform flow of other charge carriers and reduce performance of the fuel cell.

In the PEMFC, hydrogen is dissociated at anode surface and protons and electrons are created through exothermic reaction in which heat evolves. This hot surface-caused wave function for the electrons that form on it, in turn, changes with thermal energy as well. Assuming the electrons are being deaccelerated to produce the current and this deacceleration formed photon of different energy and we also assume monochromatic photon generated inside PEMFC. It has been shown in the literature that PEMFCs can achieve high levels of efficiency if efficient thermal management is established. For instance, the PEMFC conventionality efficiency is 50–60% [3], and the thermal optimization of PEMFC has been visible the way to increase efficiency up to 85% [4]. While PEMFCs are relatively of high performance with increase in temperature and pressure, yet they lack operational stability [5]. Besides, the performance of PEMFC first goes up and then down with time [6].

The investigation of scattering could provide a solution to control the heat produced in cell and enhance its performance [7]. Future developments will probably concern destacked/HT based PEMFCs, as they are more sensitive to heat capacity variation [8]. Yet, the detailed origin of heat generation and effects on temperature diminution are still not fully understood. The fuel cell performance is also significantly impacted by the temperature, leading towards different current densities and different energy losses [9,10] has theoretically investigated the inelastic scattering processes including the DCS under a weak laser field. Current work in progress includes the formation of models for the temperature dependence of DCS based on adjusted thermal Volkov wave functions [11,12]. Output current is as well part of other contributions that explore the effect of free electron-ion interactions on output current [13], and also the influence of thermal quantum species screening on anode performance [14].

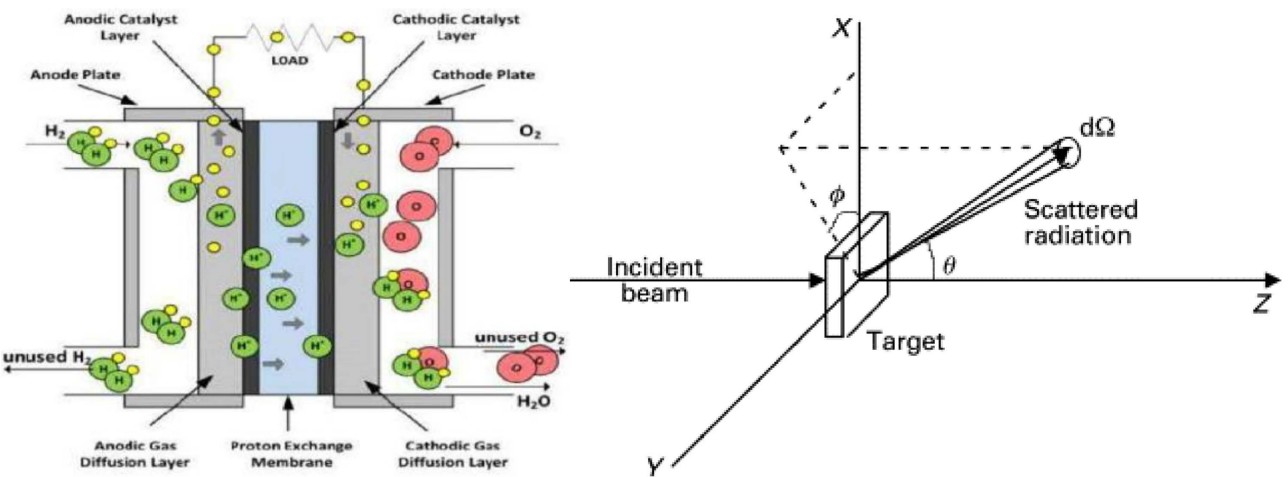

**Fig 1. PEMFC (left side) and scattering around electrode of PEMFC (right side).**

## Methods and materials

### Theoretical model

As described in Fig 1, the mechanism formation of electron and proton on the surface of electrode the wave function of electron without heat or any field is given by equation 1 as shown below,

$$\psi = Ae^{i(kr-\omega t)} \tag{1}$$

Due to the high flow of hydrogen into the anode channel of the PEMFC, the deceleration of electrons results in the emission of photons with varying energy but we considered monochromatic in this reserch work.. In our work, we model this photon field as a weak laser field, generated by the dynamic motion (acceleration and retardation) of electrons within the system. The electron is laser field define by Volkov wave function [15] as

$$X(r,\ t) = \frac{1}{(2\pi)^{3/2}} exp\left\{ i\frac{p}{h} \cdot \left( r + \frac{e}{m}\int A(t)dt \right) - i\frac{E}{\hbar}t - i\frac{e^2}{2m\hbar}\int A^2(t) \right\} \tag{2}$$

Where, Vector potentials defined as $A = a\left[\hat{x}\cos(\omega t) + \hat{y}\sin(\omega t)\tan\left(\frac{\eta}{2}\right)\right]$, and ellipticity $\eta = 0$ for linear polarization, ellipticity $\eta = \pm\frac{\pi}{2}$ for circular polarization and ellipticity $\eta \neq \pm\frac{\pi}{2}$ for ellitpical polarization, $p$ is monetum of electron, $r$ is position of scattering electrons, E is energy of electron. On using equation (2) we can formulate scattering matrix the interaction region of charge particles the surface we have S-Matrix and T-Matrix [16] as shown in equation (3)

$$S_{fi} = \delta_{fi} - \frac{i}{\hbar}\int_{-\infty}^{+\infty} \langle X_f(r,t)\,|V(r)|\,X_i(r,t)\rangle\, dt \tag{3}$$

The second term of equation (3) is also known as transition matrix (T-Matrix) which is used to calculate the scattering dynamics of electron on the surface of electrode of PEMFC by target around the electrode. Once electrons are generated at the anode surface, they are subjected to the field created by other self-generated electrons undergoing deceleration. Simultaneously, the exothermic reaction responsible for splitting hydrogen into protons and electrons continues to release heat. As a result, the electrons exist in both a laser-like (photon) field and a thermal environment at the anode surface of

the PEMFC. Therefore, the Volkov wave function must be adapted to account for thermal effects arising from exothermic reactions and collisions [10]. The wave function of equation (2) is modified as thermal Volkov wave function as $\langle X_{efT}(r, t)|$ and $|X_{eiT}(r, t)\rangle$ which represent after scattering and before scattering, respectively and defined as

$$|X_{eiT}(r,\ t)\rangle = \frac{1}{(2\pi)^{3/2}}exp\left\{i\frac{p_{eiT}}{h}.\left(r+\frac{e}{m}\frac{a}{\omega}\sin(\omega t)\right)-\frac{i}{\hbar}\left(E_{eiT}+\frac{a^2e^2}{4m}\right)t-i\frac{e^2a^2\sin(2\omega t)}{8m\hbar\omega}\right\}-k_e\nabla T_{eiT}\exp(i\omega_{eiT}t) \tag{4}$$

$$|X_{efT}(r,\ t)\rangle = \frac{1}{(2\pi)^{3/2}}exp\left\{i\frac{p_{efT}}{h}.\left(r+\frac{e}{m}\frac{a}{\omega}\sin(\omega t)\right)-\frac{i}{\hbar}\left(E_{efT}+\frac{a^2e^2}{4m}\right)t-i\frac{e^2a^2\sin(2\omega t)}{8m\hbar\omega}\right\}-k_e\nabla T_{efT}\exp(i\omega_{efT}t) \tag{5}$$

In equation (4) and (5) the last term $k_e\nabla T_e\exp(i\omega_{eiT}t)$ is thermal electron wave function generate at the surface of electrodes, where $k_e$ thermal conductivity of elections and $\nabla T_e$ is change is tetmapture of thermal electon and $\omega_{eiT}$ is frequency of electrons in thermal environment. Considering the thermal environment, the thermal Volkov wave function is defined equation (4) and (5) with thermal hydrogen-like atom with nuclear charge Z as $V(r) = \left(\frac{4\zeta(3)}{3\pi\beta^3}r^2 - \frac{4\zeta(5)}{5\pi\beta^5}r^4\right)$ [17]. By substituting the expressions of thermal Volkov wave function and potential into the integration of Equation (3), and applying the method developed by Kroll and Watson in 1973 for elliptically polarized fields, the DCS is obtained using $\frac{d\sigma}{d\Omega} = \frac{p_{efT}}{p_{eiT}}\left|T_{fi}\right|^2$ as,

$$\frac{d\sigma}{d\Omega} = \frac{p_{efT}}{p_{eiT}}\left|J_n(R\cos\gamma)f^1_{Born}+k_eT_{cell}\frac{\delta(\omega_{efT}-\omega_{eiT})}{\delta(E_{efT}-E_{eiT}+n\hbar\omega)}\right|^2 \tag{6}$$

Scattering angle $\gamma$ is defined as $tan^{-1}\left\{tan\theta.\ tan\left(\frac{\eta}{2}\right)\right\}$ and also

$$f_{Born}^{(1)} = -\frac{4\pi}{q}\left[-\frac{4\zeta(3)}{3\pi\beta^3}\left(\frac{e^{iqr}\left(-iq^3r^3+3q^2+6iqr-6\right)}{q^4}+\frac{e^{-iqr}\left(iq^3r^3+3q^2-6iqr-6\right)}{q^4}\right)-\frac{4\zeta(5)}{5\pi\beta^5}\left(\frac{e^{-iqr}(iq^5r^5+5q^4r^4-20iq^3}{\phantom{0}}\right.\right.$$
$$\tag{7}$$

The temperature considered in equation (6) arises from the exothermic reaction occurring at the anode, which also corresponds to the typical operating temperature of the PEMFC. The output current of the PEMFC can be described using the Butler–Volmer equation as,

$$I = A.i_0\left\{\exp\left[\frac{\beta_a n_e F}{RT}(E-E_{eq})\right]-\exp\left[-\frac{\beta_c n_e F}{RT}(E-E_{eq})\right]\right\} \tag{8}$$

By rearranging equation (7–8), the temperature-dependent term can be expressed as $T_{cell} = \frac{n_e F(E-E_{eq})}{R\log\left(\frac{I}{AN i_0(e^{\beta_a}-e^{\beta_c})}\right)}$, also we have $\eta = \frac{E_{cell}}{E^o}\times 100\%$ [18,19] and V = IR with modification of these for I, we have Now the equation (6) becomes,

$$\frac{d\sigma}{d\Omega} = \frac{p_{efT}}{p_{eiT}}\left|J_n(R\cos\gamma)f^1_{Born}+k_e\frac{n_e F(E-E_{eq})}{R\log\left(\frac{E_{cell}}{AR_N i_0(e^{\beta_a}-e^{\beta_c})}\right)}\frac{\delta(\omega_{efT}-\omega_{eiT})}{\delta(E_{efT}-E_{eiT}+n\hbar\omega)}\right|^2 \tag{9}$$

Scattering angle $\gamma$ is defined as $tan^{-1}\left\{tan\theta.\ tan\left(\frac{\eta}{2}\right)\right\}$. Equation (9) was computed numerical computation, the following parameter values were selected based on existing literature and physical relevance: the incidence energy was varied

from 0 to 20 eV [7], as beyond this range the behavior becomes constant and lacks significant physical interpretation. The scattering angle θ was taken from 0° to 360°, order of Bessel function first, $k_e = 1$ a.u., photon energy 1.17 eV, distance separation 0–50 nm, a = 3 a.u. [12,20], time is for pulse in picosecond range [20].

## Computation work frame with error and limitation

The computational environment has been refreshed after resetting the workspace by clearing variables with clc and close all, thus preventing interference from previous variables or plots (step 1). Important physical parameters are introduced (step 2). The resultant equation (9) is written for the established expressions coding to investigate nature of DCS (step 3). The outcome is plotted: different colours represent the orders (step 4). Labels, legends, font sizes and annotations are then added in a controlled manner to improve reading of plots and interpretation (step 5). This stepwise flow diagram is for MATLAB efficiently simulates and visualizes as is demonstrated in below (Fig 2).

The leading error in this work lies in the use of distinct assumptions from that of conventional theoretical models like WKA. The model is constructed with strict MATLAB programming that restricts the simulation and cross-validation. Moreover, the analysis is based on Bessel functions up to third order only, which may lead losses of accuracy in those cases where higher-order contributions are relevant. The work is restricted under non-relativistic conditions, and deals with both the thermal and non-thermal cases without considering some relativistic effects which may be significant in high energy devices. In addition, all spin-related factors are completely ignored so that this may result in incomplete or even wrong answer for processes with important role of spin dynamics.

## Experiential model

For experimental verification of developed equation (9) and study the behavior of the DCS under varying PEMFC parameters, an experiment was designed as shown in Fig 3. The internal component of experimental setup includes, proton exchange membrane, current collector, diffusion layer, gasket, nut, bolts, electrode with platinum coated on stainless steel, iron plated, etc. The electrode was prepared by coating stainless steel with platinum using the electroplating method

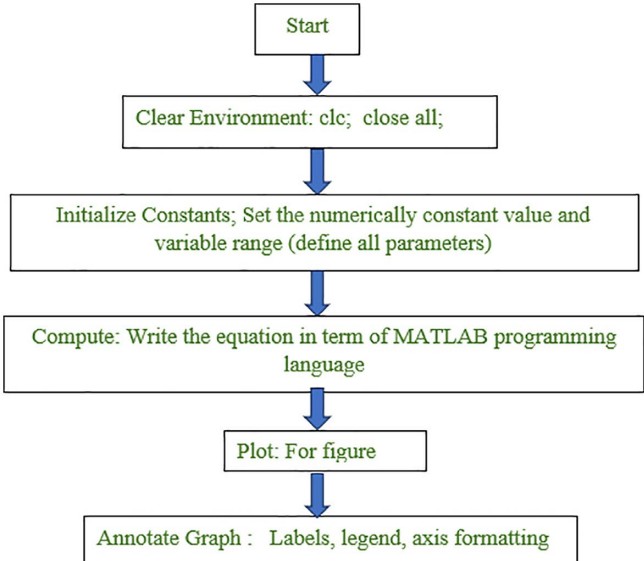

**Fig 2. Computational framework of developed equation for MATLAB.**

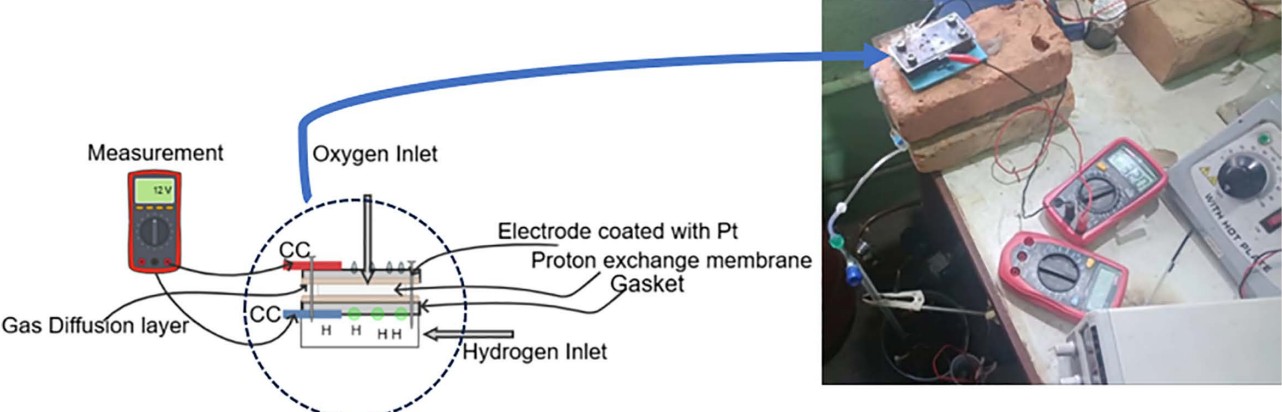

**Fig 3. PEMFC as well as electrolysis combines prototypes.**

(Aqua Regia solution was prepared to dissolved 1 mg of platinum which acts as electrolyte and coating plate is connect to negative terminal at 45 °C, 4 V for 20 seconds). On assembling the component, a porotype is developed as sketch and real experiment is shown in Fig 3 to observed the data using multimeter with supply of $H_2$ prepared by electrolysis methods. The activation area of electrode is 1.5 cm² and inlet of hydrogen is 2 mL per minutes.

## Statistical analysis

For the uncertainty study, two factors were taken into consideration (1) System stability; prototype form which data should be generated and (2) Data scatter around experimental measurement temperatures from developed prototype. $U_v$ is the uncertainty of a built setup (prototype) operating at 1 mL/min, in our case $U_t$ is the uncertainty connected with measurements performed under external heating. The thermal condition was induced and maintained by an external heat source because of the lack of internal thermo-generative ability in a prototype. The total uncertainty of the measurement is given by, $U = \sqrt{U_V^2 + U_T^2}$ and this can be used to determine the uncertainty in the measurement. Also standard error (SE) bar plot of when measurement was taken with considering the temperature defined by $SE = SD/\sqrt{n}$ and standard deviation (SD) is defined as $SD = \sqrt{\frac{\sum(x_i - \bar{x})^2}{n}}$ where $x_i$ is each indivisual measurement, $\bar{x}$ is average in measurement and n repetition of observation.

## Ethics statement

Ethical approval is not required for this work since it is a computational work and we self-designed the theoretical model. The experimental part dose not participant any human/animals trail, it is materials based where we design and experiment the system.

## Results and discussion

### Characterization of preprepared electrode

In Fig 4 show characteristic of X-ray diffraction patterns for identification and conformation of paltium coated in electrode surface of PEMFC. Sharper, more intense peaks indicate a well-crystalline material, whereas broader or less intense ones could result from smaller crystallites and/or defects/amorphous regions in the material around 20°. In particular, it is evident that the peak at 20° (2θ) shows an intensity of 3300 a.u., resulting in a well characterized crystalline phase. At 40°, the peak conforms platinum of (111) crystalline structure and at 83⁰ conform another crystalline phase conform platinum

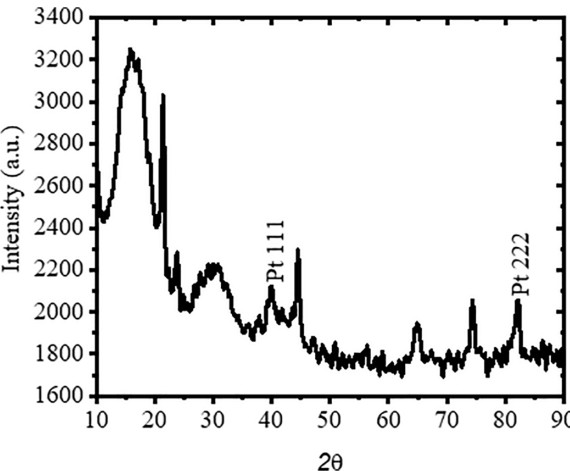

**Fig 4. XRD of electroplating of platinum on stainless steel.**

of (222) [21]. This conformation shows the deposition of Pt on electrode and hence the electrode is used for PEMFC as catalytic electrode for ionization of hydrogen.

### Effect of temperature on output voltage of PEMFC developed prototypes

After developing the prototype, a series of experiments were conducted with a hydrogen inlet rate of 2 mL per minute. Experimental observations revealed that as the temperature increased, the output voltage decreased, as shown in Fig 5(a). We considered four case 30 °C-35 °C, 35 °C −40 °C, 40 °C −45 °C and 45 °C −50 °C. The measure was conducted as with supply of 2 mL hydrogen per miunte for this we consider a heat chamber of control system whose temperature is 35 °C and PEMFC is palce on it and temperature in then decrease to 30 °C and found the with decrease in temperature voltage increase. After than the same system is cool at room temperature and then palce on heat 40 °C inside the heating chamber and cool up to 35 °C this also shows the voltage increase with decrease in temperature and similar other 40 °C −45 °C and

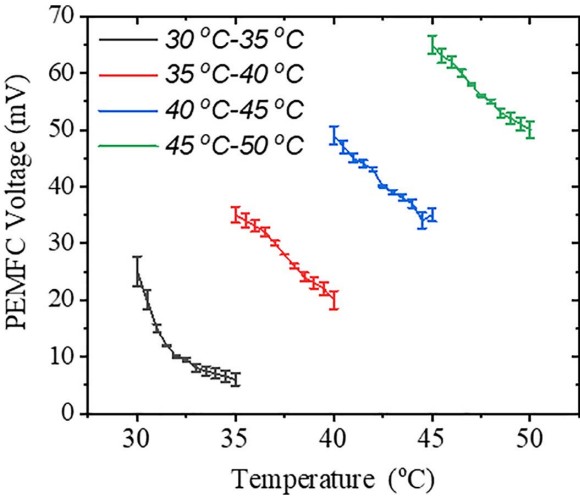

**Fig 5. Output voltage of PEMFC effected by temperature.**

45 °C −50 °C is done. This cycle was repeated multiple times. From these observations, it was concluded that the application of heat reduces the output voltage of the PEMFC. This suggests that the electric field generated by the electrons, which are formed during hydrogen supply, is influenced by temperature. A lower voltage indicates a weaker electric field. A weaker field results in a wider spatial separation of electrons at the anode surface [22], thereby enlarging the interaction region. Since the interaction region determines the probability of scattering, a larger region implies a higher likelihood of scattering, which corresponds to a larger DCS.

Without using external heating, the experiment was repeated 5 times and the measured voltage was 25 mV, with an uncertainty of 1.57 mV. After applying heat, the experiment was repeated 5 times at temperatures between 30 °C and 50 °C. The total combined uncertainty on measurement is shown in Table 1 below shows the detailed temperature-wise uncertainty values

In Fig 6(a), the DCS decreases with exchange current density of surface area electrode. The same indication can be gained in the case of a Pt/C activation potential at 60 mV as shown in Fig 6(b), but to a lesser extent than that at 50 mV. At first, the DCS rises to a maximum based on constructive interference and thereafter its value is larger, as depicted in Fig 6 (a). The further decline in DCS is due to competition between formed quantum species and charged at the electrode surface, which causes a screening impact and then decreases of the DCS. In contrast, as in Fig 6(b), the temperature

**Table 1. Uncertainty in measurement.**

| Temperature (°C) | Uncertainty in measurement (mV) | Temperature (°C) | Uncertainty in measurement (mV) | Temperature (°C) | Uncertainty in measurement (mV) | Temperature (°C) | Uncertainty in measurement (mV) |
|---|---|---|---|---|---|---|---|
| 30 | 25±1.57 | 35 | 35±1.18 | 40 | 49±1.23 | 45 | 65±1.24 |
| 30.5 | 20±1.26 | 35.5 | 34±1.14 | 40.5 | 47±1.13 | 45.5 | 63±1.14 |
| 31 | 15±1.05 | 36 | 33±1.1 | 41 | 45±1.06 | 46 | 62±1.1 |
| 31.5 | 12±1 | 36.5 | 32±1.07 | 41.5 | 44±1.04 | 46.5 | 60±1.04 |
| 32 | 10±1.01 | 37 | 30±1.02 | 42 | 43±1.02 | 47 | 58±1.01 |
| 32.5 | 9.5±1.02 | 37.5 | 28±1 | 42.5 | 40±1 | 47.5 | 56±1 |
| 33 | 8±1.05 | 38 | 26±1.01 | 43 | 39±1.02 | 48 | 55±1.01 |
| 33.5 | 7.5±1.06 | 38.5 | 24±1.06 | 43.5 | 38±1.04 | 48.5 | 53±1.06 |
| 34 | 7±1.08 | 39 | 23±1.09 | 44 | 37±1.06 | 49 | 52±1.09 |
| 34.5 | 6.5±1.1 | 39.5 | 22±1.13 | 44.5 | 34±1.18 | 49.5 | 51±1.13 |
| 35 | 6±1.11 | 40 | 20±1.22 | 45 | 35±1.13 | 50 | 50±1.17 |

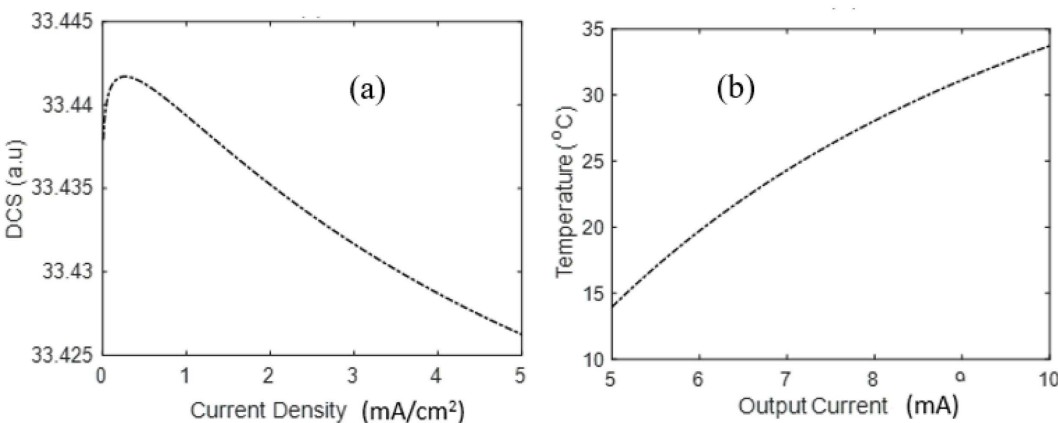

**Fig 6. Scattering dynamics of electron with (a) current density and (b) Temperature vs Output current.**

 

still increases with current density under large surface area. In the latter case, collisions further promote the exothermic energy so that temperature continues to increase. This comparative behavior highlights the role of surface area in determining the heat dynamics in PEMFCs. Fig 6 has the same nature as [20] in the model of scattering and DCS relation.

Fig 7 illustrates the behavior of the DCS as a function of scattering angle at different output voltages generated by the PEMFC. The DCS decreases and reaches a minimum around 190 degrees before increasing again with further scattering. This minimum peak near 190 degrees is attributed to destructive interference between scattering participant at the anode and around the electrode. Additionally, the DCS at 10 mV is higher than at 50 mV and 100 mV. This occurs because, at lower voltages, the electric field is weaker, resulting in greater separation between interacting particles. Conversely, at higher voltages, the DCS decreases, indicating stronger interactions and reduced separation between the particles involved. Also, Rajamani et al. [23] observed higher the voltage and stronger the electric field and vice-versa which is fit out system when temperature increase.

The DCS increases with electron incidence energy and goes through a maximum at 2.1 eV for all applied voltages. The peak is due to the constructive interference of scattering particle on the and around electrode surface. At higher energies, the DCS falls and approaches almost constant when moving beyond these values of incidence energy. This latter behavior indicates that the studied energies are not high enough to overpower the repulsive target field. However, the DCS of 10 mV is larger than those at 50 and 100 mV. Makhoute et al. [24] have studied the DCS for Helium atom and concluded that DCS increase to some extent and attain maximum value and later decrease and become constant was also found in our present work as seen from Fig 7. Also, Bartschat et al. [25] DCS associated for energy with electron Chen and found the DCS are similar. They also do not interline or construct model for PEMFC but they only explore the DCS with conventional techniques as the LASS (Fig 8).

Fig 9 indicates that the DCS increases with incident energy and is largest at 50% PEMFC efficiency, smaller at 10% PEMFC efficiency and, as we might imagine, intermediate when a value of 100% PEMFC Efficiency is chosen. It reveals that the PEMFC performance is highly dependent on DCS. Modulation of the DCS of interacting particles at the electrode surface following formation can lead to increased performance and efficiency in PEMFC. In fact, for the high efficiency, the DCS cannot be very high and also too low. The DCS relationship helps to elucidate the behaviour between particles at and around electrode surface. Moreover, the flow of hydrogen can be effectively controlled according to the active surface area of electrodes thereby enhancing the efficiency of PEMFC by adjusting interaction or scattering probability at the electrode surface. The 1.3 eV and 2.9 eV peaks

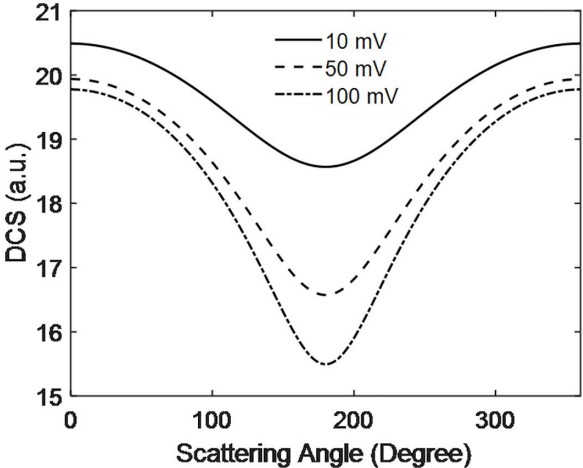

**Fig 7. Scattering dynamics of electrons with hydrogen with scattering angle at PEMFC voltage.**

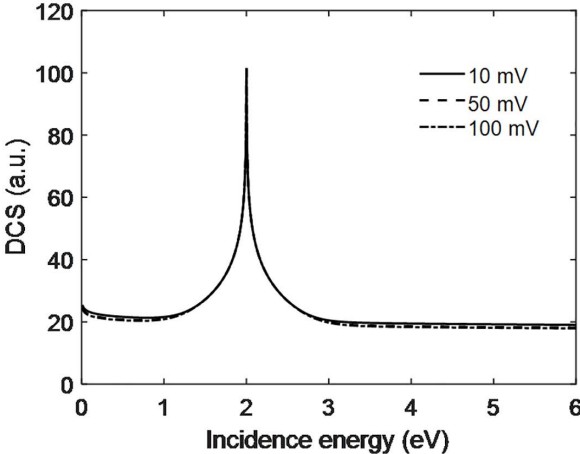

**Fig 8. Scattering dynamics with incidence energy of electron at output voltage of PEMFC.**

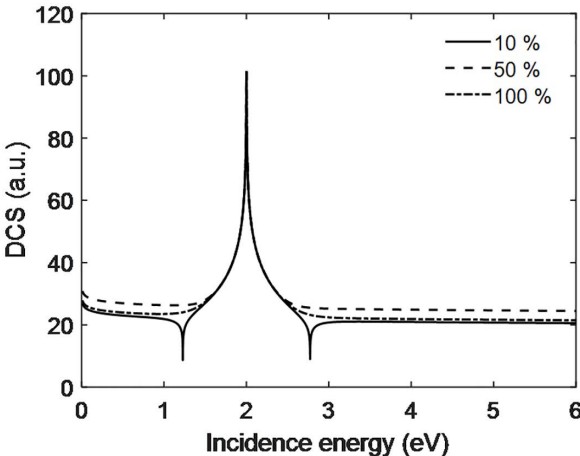

**Fig 9. Scattering dynamics with incidence energy of electron at different efficiency of PEMFC.**

appear due to destructive interference of projectile with target, and the 2.1 eV represents constructive interference. According to such performance of the DCS, PEMFCs can be developed and advanced for better utilization in practical situations. Garland et [26] have investigated differential cross section for electron impact ionization as a function of energy of the incident electron and revealed identified that DCS first go to minimum values up to certain extent and then they increases become maximum and goes to minimum value as enhance incident electrons energies.

Fig 10, shows the DCS with scattering angle for different PEMFC efficiencies. The down peaks represent the destructive interference and the up flat peaks demonstrate weak particle interactions. The distributions show that there is an intermediate range (0–100 degrees) of smaller scattering angles in which the DCS and cross section are more efficient. The DCS shows a clear destructive interference at 10% efficiency with significant magnitude, while at 100% the DCS is lower between about 100 degree and 250 degrees with two distinct destructive interference spots. On the contrary, in 50% efficiency no destructive interference is observed, but DCS values are higher than for 10 and

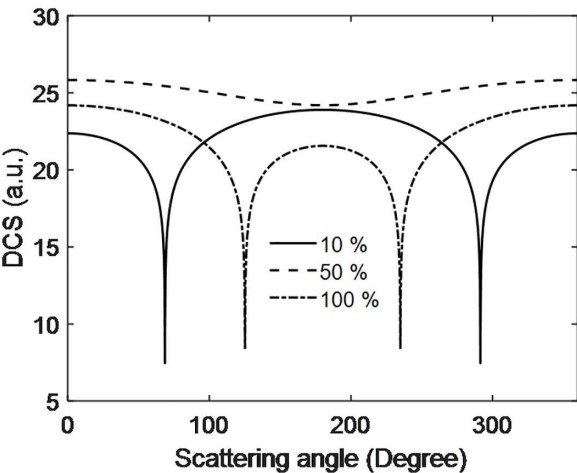

**Fig 10. Scattering dynamics of electron with scattering angle at different efficiency of PEMFC.**

100%. These findings highlight the possibility that scattering angle heavily modifies DCS at different PEMFC efficiencies. The characteristic of DCS at 50% efficiency vs scattering angle is comparable to the differential Thomson scattering cross section as a function of the scattering angle reported [27] in absence of laser. Other DCS at 10% and 100% a few more resembling of [28]. They consider electron elastic-scattering cross sections computed using two popular atomic potentials. Further they derived the model for DCS equation (9) similar to part first, however from additional term it is bottleneck of this work.

Fig 11 shows that DCS increases with the distance separation between the target and projectile particles at different PEMFC efficiencies. At 100% efficiency, the DCS exhibits intermediate values, higher than at 10% efficiency but lower than at 50%—for separations less than 2 angstroms. For separations between 2 and 5 angstroms, the DCS at 50% efficiency again plays an intermediate role, while the DCS at 10% efficiency is higher than at 50%. Additionally, destructive interference occurs across all efficiencies, indicated by deep dips in the DCS curves. Beyond a separation of 5–10

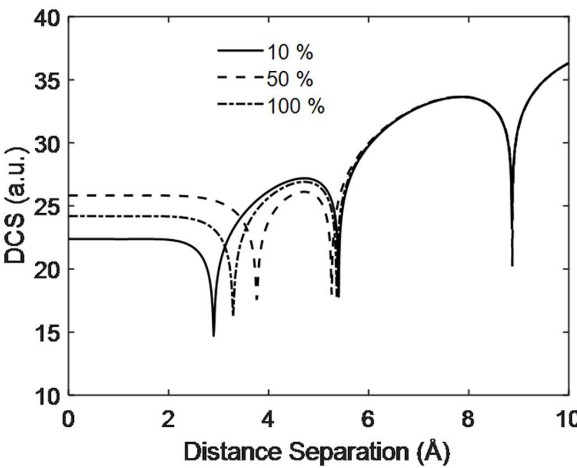

**Fig 11. Scattering dynamics of electron with distance separation at different efficiency of voltage of PEMFC.**

angstroms, another deep peak is observed, where the DCS values for all considered PEMFC efficiencies converge. This suggests that significant variation and notable effects in DCS primarily occur at lower separations. Comparing the find of DCS with distance separation of Fig 11 with Kurmi et al. [29] show the DCS increase with separation in general but they study the DCS of laser assist quantum dot scattering.

Fig 12 shows the DCS with distance separation at different PEMFC output voltages. The results indicate that the DCS at 10 mV output voltage is lower, while at 50 mV it is higher, with the 100 mV case playing an intermediate role exhibiting similar behavior up to 6 angstroms. This suggests that at smaller separations, no interference occurs, but beyond 6 angstroms, interference take place between the projectile and target particles. The deep dips in the DCS correspond to destructive interference, while the peaks are due to constructive interference between the particle participant in scattering generated inside the PEMFC. Since the electric field relates to the voltage, lower voltage corresponds to larger separation between interacting particles, which increases the interaction region and thus the probability of particle interaction (DCS). Therefore, by analyzing the relationship between output voltage and DCS, PEMFC efficiency can be optimized. This demonstrates that the performance of the PEMFC is influenced by DCS, and vice versa.

Mainly two PEMFC electrodes with Pt/C catalysts having activation potentials of 40 mV and 55 mV used in PEMFC. The nature of the DCS was studied in relation to efficiency, as shown in Fig 13. The observations indicate that both activation potentials exhibit similar behavior. Below 20% efficiency, the 40 mV activation potential dominates, while from 20% to 40%, the 55 mV potential becomes dominant. Beyond 40%, the 55 mV potential continues to dominate, although the behavior of both remains very close and similar. This suggests that the selection of activation potential, guided by DCS analysis, plays an important role in PEMFC performance. Below 50% efficiency, both destructive and constructive interference are observed, causing fluctuations and disturbances. However, beyond 50% efficiency, such interference disappears, leading to smoother PEMFC performance. Fig 13 lack limitation of exact validation but as above validation was done so this also valid because the nature of DCS was obtained form same equation (9) developed above where above result was valid

Fig 14 shows that the DCS decreases with increasing anodic charge transfer coefficient at 10% PEMFC efficiency, while it increases linearly at 100% efficiency. For 50% efficiency, the DCS initially rises and reaches a maximum due to constructive interference at a charge transfer coefficient of 0.3, then decreases beyond this point. This interference

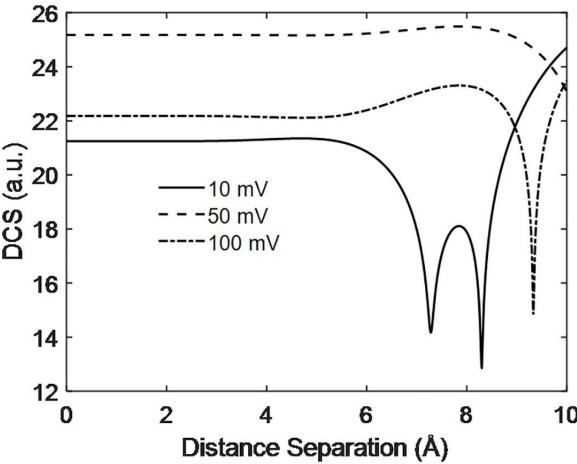

**Fig 12. Scattering dynamics with distance separation at different output voltage of PEMFC.**

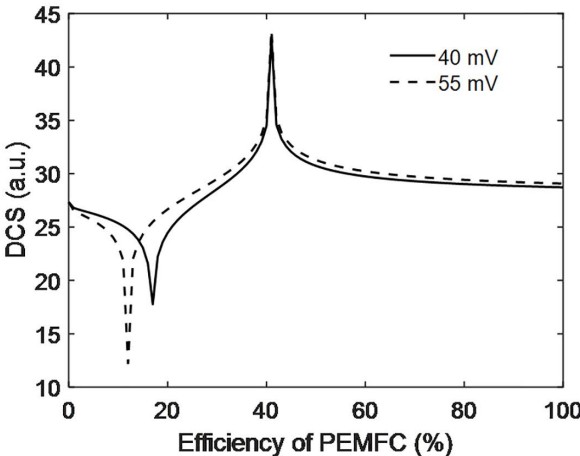

**Fig 13. DCS with efficiency of PEMFC.**

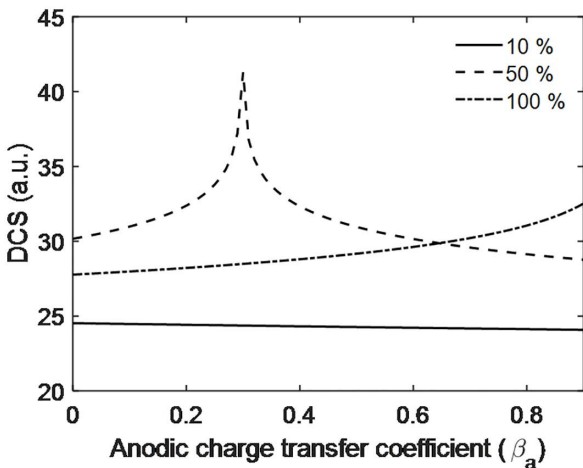

**Fig 14. Scattering dynamics with Anodic charge transfer coefficient of PEMFC electrode.**

indicates fluctuations in both the DCS and PEMFC performance. Therefore, in terms of maintaining consistent power output and uniform operation, 10% and 100% efficiencies perform better than 50%. This also suggests that the flow of charge and the separation or interaction region of charged particles remain more uniform at 10% and 100% efficiencies, whereas 50% efficiency experiences slight fluctuations. Similar to Fig 13, Fig 14 is also valid indirectly as above equation (9) is valid and Fig 14 is developed from above equation (9).

Fig 15 shows the effect of cathodic charge transfer on the DCS at different PEMFC efficiencies. The results indicate that for 50% and 100% efficiencies, the DCS decreases as the cathodic charge transfer increases, whereas for 10% efficiency, the DCS increases with increasing cathodic charge transfer. Additionally, no fluctuations are observed, suggesting the absence of constructive or destructive interference.

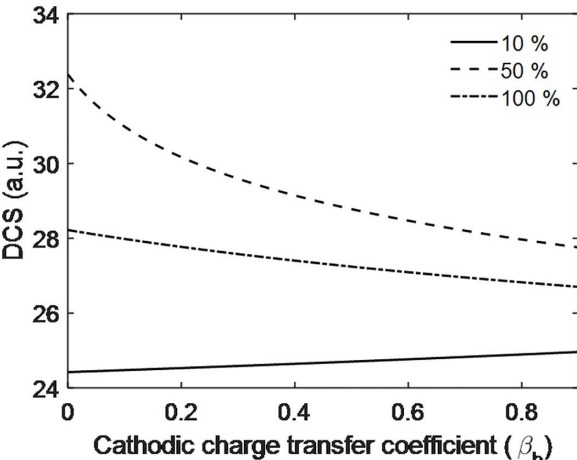

**Fig 15. Scattering dynamics with cathodic charge transfer coefficient of PEMFC electrode.**

## Conclusion

In this work, a comprehensive study of the DCS behavior in PEMFCs was presented with several factors such as output voltage, incidence energy, electrode efficiency, particle separation analysis and charge transfer coefficients. The data reveal that DCS exhibits a strong voltage and efficiency dependence, where increased efficiencies cause higher voltages to result in higher DCS as particle and phase separation become more pronounced. The PEMFC performance was affected by both constructive and negative interferences that were observed at certain energies and scattering angles, especially around 2eV and 190°. DCS behavior dependent on the efficiency indicates that optimal scattering conditions will be provided at around 50% efficiency for frequently occurring higher and lower efficiencies, which exhibits oscillation or weakened cross talk. The anodic and cathodic charge transfer coefficients also greatly influence DCS, which clearly demonstrate PEMFCs are the sensitive to electrochemical parameters. Our results are in agreement with other works (Rajamani et al., Makhoute et al. and Bartschat etal.) but go beyond other models by taking realistic PEMFC operating conditions as an input. However, efficient and upgraded performance of PEMFC could be improved by controlling DCS flow through precise control of voltage, incidence energy and electrochemical conditions. The paper highlights the significance in considering scattering mechanisms for optimisation and a potential role of DCS in PEMFC system design & control.

## Acknowledgments

The authors would like to express their sincere gratitude to all faculty members of the Department of Physics, Patan Multiple Campus, Tribhuvan University, Patan Dhoka, Lalitpur, Nepal, for providing a peaceful and supportive environment for both experimental and theoretical work. We also extend our appreciation to the Central Department of Physics, Tribhuvan University, Kirtipur, Nepal, for their valuable support and guidance from various perspectives.

## Author contributions

**Conceptualization:** Ajay Kumar Jha.

**Formal analysis:** Suresh Prasad Gupta, Ajay Kumar Jha.

**Investigation:** Saddam Husain Dhobi.

**Methodology:** Saddam Husain Dhobi.

**Resources:** Suresh Prasad Gupta.

**Software:** Ajay Kumar Jha.

**Supervision:** Kishori Yadav, Jeevan Jyoti Nakarmi, Ajay Kumar Jha.

**Validation:** Saddam Husain Dhobi, Kishori Yadav, Suresh Prasad Gupta, Jeevan Jyoti Nakarmi, Ajay Kumar Jha.

**Visualization:** Saddam Husain Dhobi, Suresh Prasad Gupta, Jeevan Jyoti Nakarmi, Ajay Kumar Jha.

**Writing – original draft:** Saddam Husain Dhobi.

**Writing – review & editing:** Kishori Yadav, Suresh Prasad Gupta, Jeevan Jyoti Nakarmi, Ajay Kumar Jha.

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
