## [Decision Letter · Decision Letter 0]

12 Nov 2025

Dear Dr. Dhobi,

Thank you for submitting your manuscript to PLOS ONE. After careful consideration, we feel that it has merit but does not fully meet PLOS ONE’s publication criteria as it currently stands. Therefore, we invite you to submit a revised version of the manuscript that addresses the points raised during the review process.

We look forward to receiving your revised manuscript.

Kind regards,

Narayanamoorthy Bhuvanendran, Ph.D.

Academic Editor

PLOS ONE

Journal Requirements:

“UGC-Nepal”

“Authors has no competing interest”

6. Please provide a complete Data Availability Statement in the submission form, ensuring you include all necessary access information or a reason for why you are unable to make your data freely accessible. If your research concerns only data provided within your submission, please write "All data are in the manuscript and/or supporting information files" as your Data Availability Statement.

Reviewers' comments:

Reviewer's Responses to Questions

**Comments to the Author**

1. Is the manuscript technically sound, and do the data support the conclusions?

Reviewer #1: Yes

Reviewer #2: Partly

2. Has the statistical analysis been performed appropriately and rigorously?

Reviewer #1: Yes

Reviewer #2: No

3. Have the authors made all data underlying the findings in their manuscript fully available?

Reviewer #1: Yes

Reviewer #2: Yes

4. Is the manuscript presented in an intelligible fashion and written in standard English?

Reviewer #1: Yes

Reviewer #2: Yes

Reviewer #1: The paper concludes that external temperature negatively affects PEMFC output due to increased particle resistance and entropy, emphasizing the need for precise thermal management to improve fuel cell efficiency. However�the experimental setup is too simple. It is recommended to upgrade the experimental conditions to ensure the authenticity of the experiment.

Reviewer #2: This is a statistical review on a Research Article manuscript for PLOS ONE entitled "Temperature Dependence Laser-Assist Scattering and Its Impact on Efficiency on with PEMFC" (PONE-D-25-41737).

This paper is primarily a theoretical modeling and descriptive experimental study rather than one based on inferential statistics. The authors develop a theoretical model and then conduct a single prototype experiment to observe a phenomenon. As a result, it lacks formal statistical components.

The statistical methods used are limited to descriptive statistics.

There are no statistical experimental design considerations, no statistical modeling (or use of statistical models) and no inference (such as tests of statistical hypotheses).

The study uses a simple, non-replicated prototype design. A single PEMFC prototype was built and tested. There is no statistical sampling. The "sample" is a single, custom-fabricated device. The paper does not mention any replication of the experiment.

From the statistics standpoint, the paper fails to deliver on the following important rubric of PLOS ONE:

"3. Experiments, statistics, and other analyses are performed to a high technical standard and are described in sufficient detail."

The above evaluation considers only statistical aspects of the paper as part of the requested statistical review of the manuscript. It does not reflect on the overall scientific merits of the presented research.

**Do you want your identity to be public for this peer review?** For information about this choice, including consent withdrawal, please see our Privacy Policy

Reviewer #1: No

Reviewer #2: No

---

## [Author Response · Author response to Decision Letter 1]

10 Dec 2025

Respond to reviewer are attached to other file in file attachment section.

---

## [Decision Letter · Decision Letter 1]

23 Dec 2025

Temperature Dependence Laser-Assist Scattering and Its Impact on Efficiency of  PEMFC

PONE-D-25-41737R1

Dear Dr. Saddam Husain Dhobi,

We’re pleased to inform you that your manuscript has been judged scientifically suitable for publication and will be formally accepted for publication once it meets all outstanding technical requirements.

Within one week, you’ll receive an e-mail detailing the required amendments. When these have been addressed, you’ll receive a formal acceptance letter, and your manuscript will be scheduled for publication.

Kind regards,

Narayanamoorthy Bhuvanendran, Ph.D.

Academic Editor

PLOS One

Additional Editor Comments (optional):

Reviewers' comments:

Reviewer's Responses to Questions

**Comments to the Author**

Reviewer #1: (No Response)

2. Is the manuscript technically sound, and do the data support the conclusions?

Reviewer #1: Yes

3. Has the statistical analysis been performed appropriately and rigorously?

Reviewer #1: Yes

4. Have the authors made all data underlying the findings in their manuscript fully available?

Reviewer #1: Yes

5. Is the manuscript presented in an intelligible fashion and written in standard English?

Reviewer #1: Yes

Reviewer #1: (No Response)

**Do you want your identity to be public for this peer review?** For information about this choice, including consent withdrawal, please see our Privacy Policy

Reviewer #1: No

---

## [Editor Report · Acceptance letter]

PONE-D-25-41737R1

PLOS One

Dear Dr. Dhobi,

I'm pleased to inform you that your manuscript has been deemed suitable for publication in PLOS One. Congratulations! Your manuscript is now being handed over to our production team.

Kind regards,

on behalf of

Dr. Narayanamoorthy Bhuvanendran

Academic Editor

PLOS One